# Association between Traumatic Subarachnoid Hemorrhage and Acute Respiratory Failure in Moderate-to-Severe Traumatic Brain Injury Patients

**DOI:** 10.3390/jcm11143995

**Published:** 2022-07-10

**Authors:** Min Li, Rui Wang, Qi-Xing Fang, Yi-Xuan He, Ying-Wu Shi, Shun-Nan Ge, Rui-Na Ma, Yan Qu

**Affiliations:** 1Neurocritical Care Unit, Department of Neurosurgery, The Second Affiliated Hospitals, Fourth Military Medical University, Xi’an 710038, China; neursylm@fmmu.edu.cn (M.L.); drmedfang@163.com (Q.-X.F.); hyxzlzy@163.com (Y.-X.H.); shiyw0729@126.com (Y.-W.S.); gesn8561@fmmu.edu.cn (S.-N.G.); 2Department of Medical Management, The Second Affiliated Hospitals, Fourth Military Medical University, Xi’an 710038, China; tdjrjz@fmmu.edu.cn; 3Department of Pulmonary and Critical Care Medicine, The Second Affiliated Hospitals, Fourth Military Medical University, Xi’an 710038, China; maruina84@fmmu.edu.cn

**Keywords:** traumatic brain injury, Fisher grade, subarachnoid hemorrhage, acute respiratory failure

## Abstract

Acute respiratory failure (ARF) with a high incidence among moderate-to-severe traumatic brain injury (M-STBI) patients plays a pivotal role in worsening neurological outcomes. Traumatic subarachnoid hemorrhage (tSAH) is highly prevalent in M-STBI, which is associated with significant adverse outcomes. In this retrospective cohort study, we aimed to explore the association between the severity of the tSAH and ARF in the M-STBI population. A total of 771 subjects were reviewed. Clinical and neuroimaging data of M-STBI patients were retrospectively collected, and ARF was ascertained retrospectively based on their electronic medical record. The degree of tSAH was classified according to Fisher’s criteria, and the grade of tSAH was dichotomized to a low Fisher grade (Fisher grade 1–2) and a high Fisher grade (Fisher grade 3–4). After exclusion procedures, the data of 695 M-STBI patients were analyzed. A total of 284 (30.8%) had a high Fisher grade on admission. The overall rate of ARF within 48 h upon admission was 34.4% (239/695); it was 29.5% (142/481) and 46.3% (99/214) for the low and high Fisher groups, respectively. In a full cohort, a high Fisher grade was associated with ARF after adjusting for age, gender, GCS, smoking history, comorbidities, multiple injuries, characteristics of TBI, and pulmonary factors (OR 1.78; 95% CI, 1.11–2.85, *p* = 0.016). This result remained robust in the comparisons after PSM (71/132, 42.8% vs. 53/132, 31.9%; OR, 1.59; 95% CI, 1.02–2.49, *p* = 0.042). A high Fisher SAH grade exposure on admission is associated with ARF in M-STBI patients.

## 1. Introduction

Traumatic brain injury (TBI) constitutes the main cause of mortality and morbidity in young patients [1]. Acute respiratory failure (ARF) is a critical early complication in patients with moderate–severe traumatic brain injury (M-STBI), which is significantly associated with the global severity of the TBI [2]. ARF with a high incidence among M-STBI patients plays a pivotal role in the poor outcomes of TBIs because a traumatized brain is extremely susceptible to hypoxia-induced injury [3,4,5]. The risk factors for ARF post M-STBI are usually related to pulmonary events such as aspiration, lung consolidation, pulmonary edema, etc. Moreover, brain–lung interactions also play an important role in ARF post M-STBI. However, the quantitative markers of brain injury for ARF are unclear.

Traumatic subarachnoid hemorrhage (tSAH) is highly prevalent (33–60%) in M-STBI, which is associated with significant mortality and adverse outcomes [6,7,8]. Thus, traumatic SAH has been shown to be a marker for the increased severity of a TBI as opposed to being an independent prognostic variable. Moreover, the extent of an SAH on an admitting CT is graded according to the Fisher scale [9]. It is well known that a higher Fisher grade of SAH correlates with complication and worse clinical outcomes in aneurysmal SAH patients [8,10,11,12]. However, the association between a higher Fisher grade of traumatic SAH and ARF is not clear.

In this study, we sought to explore the association between a high Fisher grade of traumatic SAH and ARF in the M-STBI population, hypothesizing a positive association between a high Fisher grade and the risk of ARF. This investigation explored the association between brain injury and lung injury mediated by traumatic SAH.

## 2. Materials and Methods

### 2.1. Study Design, Population, and Setting

The retrospective cohort study included adult patients (age > 18 years) with M-STBI from 21 November 2013 to 16 April 2019, at an academic hospital, Fourth Military Medical University in Xi’an. Inpatient data were obtained from the electronic health record (EHR) of the health care system.

The inclusion criteria of patients were a definite history of acute brain trauma and Glasgow Coma Scale (GCS) scores of 3–12 [13]. The following were exclusion criteria: (1) age < 18 years old or age > 80; (2) brain trauma secondary to stroke, intracranial hemorrhage, seizure, or other primary cerebral disease; (3) patients with a history of brain surgery before admission; (4) TBI with chest trauma; (5) pregnancy.

This retrospective study was approved by the Institutional Review Board (IRB) of the Second Affiliated hospital, Fourth Military Medical University (TDLL-KY-202104-03), and informed consent was waived because data were deidentified. All reporting followed the Strengthening the Reporting of Observational Studies in Epidemiology (STROBE) guidelines [14].

### 2.2. Data Collection and Primary Outcome

The medical records of the patients were carefully reviewed by five authors on separate occasions, investigating demographic parameters, symptoms, and neuroimaging features based on non-enhanced CT (epidural hemorrhage, subdural hemorrhage, brain contusion, intraventricular hemorrhage, midline shift, and degree of compression of the basal cisterns). The primary outcome of the study was ARF within 48 h upon admission, which was defined as respiratory failure with arterial partial pressure of O_2_ (aPO_2_) < 60 mmHg, either PaCO_2_ > 45 or <35 mmHg, and respiratory rate > 30 breaths/minute, or respiratory distress for at least 5 min [15]. The primary outcome was ARF, which was retrospectively evaluated based on arterial blood gas analysis (ABG).

### 2.3. Exposure and Covariates

The Fisher grade of traumatic SAH was the primary exposure of interest. The grade of traumatic SAH was evaluated based on the Fisher Grading Score [16] (see Appendix A for the details of score points). A low Fisher grade of SAH was defined as a Fisher grade ≤ 2, and a high grade was defined as a Fisher grading score ≥ 3. Other covariates included age, gender, GCS, multiple injury, comorbidities (hypertension, coronary heart disease, diabetes, chronic obstructive pulmonary diseases, stroke), neuroimaging features of TBI, medical management (venous infusion > 3000 mL before ARF, general anesthesia and operation before ARF), and pulmonary events (aspiration, intubation, lung consolidation). The characteristics of TBI were independently evaluated by two experienced neurosurgeons who were blinded to the primary endpoint. The basal cistern scale (BCS) was defined based on the degree of compression of the basal cisterns (see Appendix A for the details of BCS) [17].

### 2.4. Statistical Methods

We conducted the analysis with the following steps: (1) comparison of descriptive data from patients with and without exposure to high grade of SAH; (2) determination of risk-adjusted estimates for ARF using propensity score matching; (3) performance of sensitivity analyses to test the heterogeneity of ARF, understand potential unmeasured confounding, and assess the robustness of the findings in the primary analysis.

Continuous variables are expressed as the mean (SD) or median (interquartile range (IQR)), while categorical variables are expressed as numbers (%). Demographic characteristics of participants between low grades and high grades of SAH were compared by the Mann–Whitney U test or the chi-squared test.

Multivariate-adjusted models were used to analyze the association of a high grade of SAH with ARF. We tested different models by adjusting different risk factors step by step. The crude model was unadjusted; the adjusted model was further adjusted for age, gender, GCS, multiple injury, comorbidities (hypertension, coronary heart disease, diabetes, chronic obstructive pulmonary diseases, stroke), characteristics of TBI (epidural hemorrhage, subdural hemorrhage, brain contusion, intraventricular hemorrhage, midline shift, degree of compression of the basal cisterns), medical management (venous infusion > 3000 mL before ARF, general anesthesia and operation before ARF), and pulmonary events (aspiration, intubation, lung consolidation).

All the analyses were performed using the statistical software packages R (http://www.R-project.org (accessed on 3 April 2021), The R Foundation, Vienna, Austria) and Free Statistics software versions 1.5 (Beijing, China). A *p*-value of <0.05 was considered statistically significant.

### 2.5. Sensitivity Analysis

To test the robustness of our findings, propensity score matching (PSM) was performed with a nearest neighbor matching algorithm (1:1) and a caliper width (0.01). The variables selected to generate the propensity score were as shown in Appendix A. The PSM degree was estimated by a standardized mean difference (SMD) (Appendix A). SMD < 0.1 was considered acceptable [18]. It was indispensable to calculate the odds ratio (OR) for ARF, and a univariable logistic regression model with the robust variance estimator was applicable. Using the estimated propensity scores as weights, an inverse probability weighting (IPW) model was used to generate a weighted cohort [19]. Moreover, subgroup analyses were performed based on age, gender, moderate or severe TBI patients, and isolated head trauma or multiple injuries with head trauma to show the association between a high grade of SAH and ARF in different populations.

## 3. Results

### 3.1. Population

A total of 695 participants were included in the present analyses. The results of the exclusion process, summarizing the exclusion reasons, are shown in Figure 1.

In the high-Fisher-grade cohort of 214 participants, the median age was 53 years old (IQR, 43–60 years), and 171 (79.9%) were male. Hypertension, diabetes, and cardiovascular disease were the most common comorbidities in both groups. Patients exposed to a low Fisher grade and high Fisher grade were similar with regard to age, gender, GCS, smoking history, and comorbidities. The occurrence of aspiration, lung consolidation, and extradural hematoma (EDH) did not differ significantly between the two groups. The two groups had similar characteristics regarding the intubation and surgical procedures (decompressive craniectomy or external ventricular drain age) for intracranial hypertension before ARF.

### 3.2. Baseline Characteristics

Compared to the low Fisher grade, high-Fisher-grade patients had a significantly higher propensity for multiple injuries (*p =* 0.014), subdural hemorrhage (*p* < 0.001), brain contusions (*p* < 0.001), midline shift (MLS) (0.5 ± 0.6 vs. 0.3 ± 0.6, *p* < 0.001), compressed basal cisterns (*p* < 0.001), traumatic intraventricular hemorrhage (tIVH) (*p* < 0.001), and input greater than 3000 mL per day (*p* = 0.002) by univariate analysis. (Table 1).

### 3.3. Primary Outcome

The occurrence rates of ARF were 29.5% and 46.3% for the low- and high-Fisher-grade groups, respectively (Table 2). The propensity score-matched ARF rates for the low- and high-Fisher-grade groups were 31.9% and 42.8%, respectively (Appendix A). Exposure to a high Fisher grade of SAH was significantly associated with ARF in the entire cohort with an unadjusted OR of 2.06 (*p* < 0.001, Table 2).

### 3.4. Propensity Score Matching for Outcomes

Among 332 matched patients, 124 (37.3%) patients developed ARF. In the matched cohort, patients showing a high Fisher grade of SAH on admission had an increased risk of ARF with an OR of 1.59 (95% CI [1.02~2.49], *p* = 0.042, Table 2). IPW also demonstrated a significantly higher rate of ARF with a high Fisher grade of SAH. The OR was 1.48 (95% CI, 1.05–2.06, *p* < 0.023, Table 2).

### 3.5. Sensitive Analysis

In the full cohort, multivariate logistic regression analysis demonstrated that a high Fisher grade was associated with an increased risk for ARF after adjusting for age, gender, GCS, smoking history, comorbidities, multiple injuries, characteristics of TBI, and pulmonary risk factors in the entire cohort (Table 2). Although subgroup analysis was performed according to confounders including age, gender, GCS, and multiple injury (Figure 2), we did not observe any significant interactions in the subgroups (*p*-value for interaction >0.05 for all).

## 4. Discussion

The aim of the study was to explore the relationship between severe traumatic SAH and the occurrence of ARF within 48 h upon admission in an M-STBI population. Compared with a low Fisher grade, a high Fisher grade of traumatic SAH was independently associated with an increased risk of ARF. This result remained robust in additional models and in the comparisons after PSM. The study suggested that neurotrauma physicians need to pay attention to patients with a high Fisher grade of traumatic SAH who have priority to receive intensive airway management. This would help us to allocate resources efficiently and improve morbidity reduction by appropriately monitoring patients at risk of ARF.

There have been several clinical studies involving the risk factors of ARF post M-STBI. A retrospective cohort study by Fred Rincon et al. involved 987,305 TBI patients in the United States from 1988 to 2008 and found a positive association between several risk factors (including demographics, comorbidities, and hospital complications) and ARF [20]. However, the study did not demonstrate an association between traumatic SAH and ARF. Another study included traumatic SAH in 485 mild TBI patients and found that isolated SAH was not associated with further neurosurgical issues and clinical deterioration, but the study did not include moderate-to-severe TBI patients [7]. Bratton et al. studied ARF in isolated M-STBI patients based on the Traumatic Coma Data Bank and showed that midline shift had a 10-fold increased risk for ARF among severe TBI patients (GCS ≤ 8), but the study was not related to the association between traumatic SAH and ARF [3]. The present study fills this gap.

The pathogenesis of the association between severe traumatic SAH (Fisher grade 3–4) and ARF remains unclear in M-STBI patients. We speculated that there were three possible underlying mechanisms. The first postulation was that severe traumatic SAHs detected on initial non-contrast CT were indirect signs of diffused brain injury [17], which probably impaired the respiratory centers, thus causing ARF within 48 h upon admission. In the mild TBI population, Minoru found that massive traumatic SAH in the basal subarachnoid cisterns is not necessarily associated with parenchymal injury of the brainstem but rather with severe brainstem dysfunction. However, the study only included 20 mild TBI patients, and the conclusion was not verified by a robust statistical model [21]. In our M-STBI cohort, severe traumatic SAH was associated with ARF verified by robust logistical models and associated with poor outcomes [7,11,22,23]. Thus, severe traumatic SAH may be seen as an early marker of severer initial brain damage. The second mechanism is probably caused by a catecholamine surge. Catecholamine surge is a sudden and dramatic increase in serum levels of the catecholamines adrenaline and noradrenaline and causes the vasoconstriction of peripheral vessels, which may eventually lead to neurogenic pulmonary edema, resulting in ARF [2,24,25,26,27,28,29,30]. It often occurs if the hypothalamic–pituitary axis is injured post diffused brain injury indicated by severe traumatic SAH. The third mechanism may be due to some special molecules such as extracellular vesicles [31] induced by M-STBI, which are released into the bloody cerebrospinal fluid, then enter into systematic circulation and cause lung injury. This hypothesis warrants further investigation.

This current study had several strengths. First, we provided unequivocal epidemiological evidence for the significant correlation of severe traumatic SAH with the occurrence of ARF from a large study population. Moreover, except for the common confounders (age, gender, smoking, and comorbidities), we analyzed neurological covariates that may have effects on the development of ARF, such as GCS at admission, midline shift, compression or effacement of basal cisterns, massive lesions, and pulmonary covariates (aspiration, massive fluid administration, intubation, and cardiovascular disease). Thirdly, propensity score matching and sensitive analysis were performed to ascertain that the results were reliable.

There were also several limitations for the present study. First, due to the nature of the retrospective study design, causality could not be inferred between a high Fisher grade of SAH and ARF. Second, the study only focused on ARF within 48 h of admission, and further analysis of the dynamic respiratory dysfunction might be needed in the future. Finally, although the multiple regression models included a broad set of covariates of additional factors, some unmeasured or potential confounders/covariates might have also played a role to some extent.

## 5. Conclusions

Our findings suggested that a high Fisher grade of traumatic SAH was associated with higher risk-adjusted ARF in moderate-to-severe TBI patients. This association warrants further investigation.

## Figures and Tables

**Figure 1 jcm-11-03995-f001:**
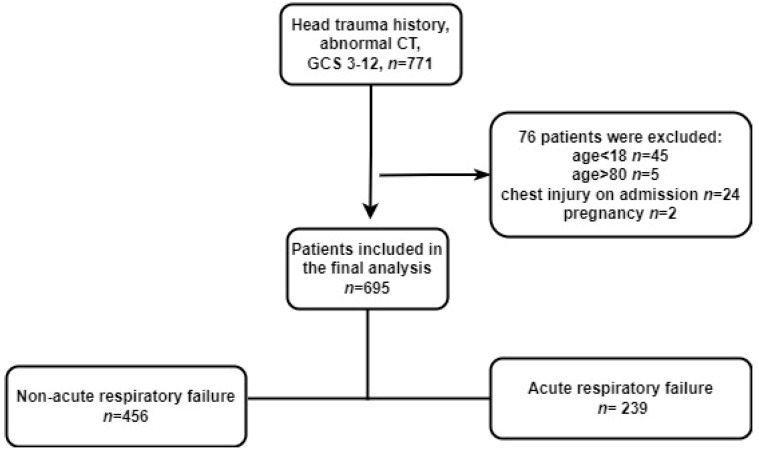
Flowchart showing the population selection procedure.

**Figure 2 jcm-11-03995-f002:**
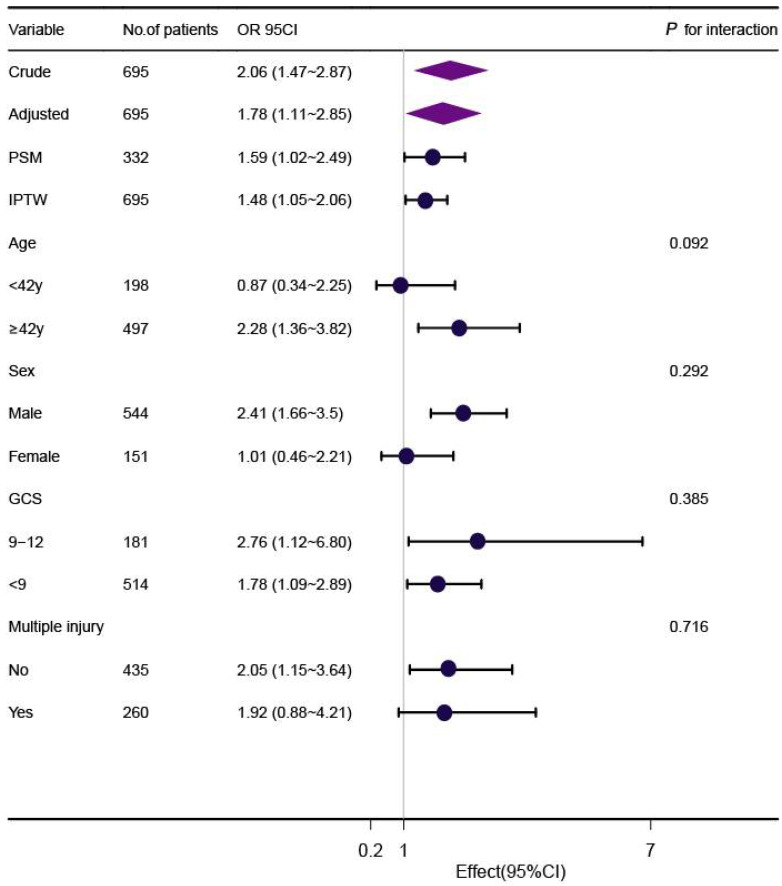
Association between high Fisher grade of traumatic SAH and ARF according to baseline characteristics. Each stratification adjusted for age, gender, GCS, multiple injury, comorbidities (hypertension, coronary heart disease, diabetes, chronic obstructive pulmonary diseases, stroke), characteristics of TBI (epidural hemorrhage, subdural hemorrhage, brain contusion, intraventricular hemorrhage, midline shift, degree of compression of the basal cisterns), medical management (venous infusion > 3000 mL before ARF, general anesthesia and operation before ARF), and pulmonary events (aspiration, intubation, lung consolidation), except for the stratification factor itself. IPTW: inverse probability weighting; PSM: propensity score-matched.

**Table 1 jcm-11-03995-t001:** Study participants’ clinical characteristics in full cohort (*n* = 695) and after propensity score matching (*n* = 332).

	Unmatched			Matched (1:1)		
Item	Low Grade	High Grade	*p*-Value	Low Grade	High Grade	*p*-Value
*n*	481	214		166	166	
Age	50 (39, 60)	53 (43, 60)	0.052	53 (44, 62)	52 (43, 60)	0.276
Male	373 (77.5)	171 (79.9)	0.551	124 (74.7)	134 (80.7)	0.235
GCS	6 (4, 9)	7 (5, 9)	0.284	7 (4, 9)	6.5 (5.0, 9.0)	0.595
Multiple injury	165 (34.3)	95 (44.4)	0.014	74 (44.6)	63 (38.0)	0.265
Smoke	175 (36.4)	83 (38.8)	0.603	56 (33.7)	60 (36.1)	0.730
HTN	52 (10.8)	27 (12.6)	0.573	20 (12.0)	16 (9.6)	0.596
DM	11 (2.3)	5 (2.3)	1.000	6 (3.6)	4 (2.4)	0.748
COPD	1 (0.2)	0 (0.0)	1.000	0 (0.0)	0 (0.0)	1.000
CAD	8 (1.7)	4 (1.9)	1.000	2 (1.2)	4 (2.4)	0.685
EDH	96 (20.0)	56 (26.2)	0.084	45 (27.1)	36 (21.7)	0.307
SDH	126 (26.2)	114 (53.3)	<0.001	82 (49.4)	78 (47.0)	0.742
Brain contusion	294 (61.1)	170 (79.4)	<0.001	121 (72.9)	132 (79.5)	0.197
IVH	56 (11.6)	90 (42.1)	<0.001	49 (29.5)	52 (31.3)	0.811
MLS	0.0 (0.0, 0.5)	0.4 (0.0, 0.8)	<0.001	0.50 (0.79)	0.53 (0.66)	0.76
Basal cistern			<0.001			0.854
0	364 (75.7)	113 (52.8)		103 (62.0)	99 (59.6)	
1	77 (16.0)	44 (20.6)		32 (19.3)	36 (21.7)	
2	40 (8.3)	57 (26.6)		31 (18.7)	31 (18.7)	
Aspiration	338 (70.3)	145 (67.8)	0.565	112 (67.5)	113 (68.1)	1.000
Intubation	241 (50.1)	119 (55.6)	0.208	87 (52.4)	91 (54.8)	0.741
Lung consolidation	135 (28.1)	63 (29.4)	0.780	44 (26.5)	45 (27.1)	1.000
Venous infusion > 3000 mL	179 (37.2)	107 (50.0)	0.002	79 (47.6)	79 (47.6)	1.000
General anesthesia and operation before ARF	434 (90.2)	199 (93.0)	0.301	149 (89.8)	153 (92.2)	0.566

Data are presented as median (interquartile range) or number (percentage). GCS: Glasgow Coma Scale; CAD: coronary heart disease; HTN: hypertension; DM: diabetes mellitus; COPD: chronic obstructive pulmonary disease; EDH: epidural hematoma; SDH: subdural hematoma; IVH: intraventricular hemorrhage; MLS: midline shift; ARF: acute respiratory failure.

**Table 2 jcm-11-03995-t002:** Associations between metformin use and outcome in the crude analysis, multivariable analysis, and propensity score analyses.

Variable	*n*	ARF (*n*/%)	Odds Ratio (95% CI)	*p*-Value
No. of events/no. of patients at risk (%)				
Low Fisher grade	481	142 (29.5)		
High Fisher grade	214	99 (46.3)		
Crude analysis	695	241 (34.7)	2.06 (1.47~2.87)	<0.001
Multivariable analysis ^a^	695	241 (34.7)	1.78 (1.11~2.85)	0.016
With matching ^b^	332	124 (37.3)	1.59 (1.02~2.49)	0.042
With inverse probability weighting ^c^	695	241 (34.7)	1.48 (1.05~2.06)	0.023

^a^ Shown is the hazard ratio from the multivariable logistic regression model, adjusted for all covariates in Table 1. ^b^ Shown is the odds ratio from a multivariable logistic regression model with the same strata and covariates with matching according to the propensity score. The analysis included 332 patients (116 exposed to high Fisher grade of SAH and 116 exposed to low Fisher grade). ^c^ Shown is the primary analysis with a hazard ratio from the multivariable logistic regression model with the same strata and covariates with inverse probability weighting according to the propensity score. ARF: acute respiratory failure.

## Data Availability

The datasets generated during and/or analysed during the current study are available from the corresponding author on reasonable request.

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
