# Peer review of "Association between Traumatic Subarachnoid Hemorrhage and Acute Respiratory Failure in Moderate-to-Severe Traumatic Brain Injury Patients"

_jcm, 2022, doi:10.3390/jcm11143995_

Round 1

Reviewer 1 Report

The authors here present a retrospective analysis of the association of the degree of a patient's traumatic subarachnoid burden with respiratory failure. This is a single-institution study with IRB approval. The authors used multivariate analysis to tease out the relationship between the Fisher score and respiratory failure within 48hrs. 

There are some concerns here with the methodology. The authors chose a Fisher score, which is typically used for non-traumatic aneurysmal disease rather than the modified Fisher score which has at least been used previously (see Sagher et al 2019, as well as others). It may be that using a scale which has not been validated for traumatic SAH might play a role in the outcomes. 

There is a well-documented association between SAH (non-traumatic) and ARF which is not mentioned in the introduction (Kahn et al 2006). 

Lastly, there is no mention in the discussion about why this finding may be clinically relevant. Patients with severe TBI will likely have significantly decreased GCS and all of the other covariates mentioned in the study. It is unlikely that they would be observed in situations outside of an ICU and their potential for intubation would be very high regardless. It is unclear how this finding might change management of patients, especially when this matric (Fisher's) is not typically used in this patient population.

Author Response

Point 1: There are some concerns here with the methodology. The authors chose a Fisher score, which is typically used for non-traumatic aneurysmal disease rather than the modified Fisher score which has at least been used previously (see Sagher et al 2019, as well as others). It may be that using a scale that has not been validated for traumatic SAH might play a role in the outcomes. 

Response 1: Thanks for the reviewer’s constructive comments. We absolutely agree that the modified Fisher score can be used to assess the traumatic SAH. However, the Fisher score was also used to assess the traumatic SAH in the TBI study as below:

Albertine P and et.al. The clinical significance of small subarachnoid hemorrhages. Emerg Radiol. 2016 Jun;23(3):207-11.

We also retrospectively assessed the modified Fisher score in our TBI cohort and analyzed the association between high grading of the modified Fisher score (modified Fisher score>2) and ARF. The result is similar to Fisher scores (Table R1). It is demonstrated that the association between severe traumatic SAH and ARF was robust.

Table R1 The association between modified Fisher score and ARF

Variable

N

ARF (N/%)

Odds Ratio(95CI)

P value

No. of events/no. of patients at risk (%)

   Low Fisher grade

481

142 (29.5)

   High Fisher grade

214

99 (46.3)

Crude analysis

695

241 (34.7)

2.06 (1.47~2.87)

<0.001

Multivariable analysis

695

241 (34.7)

1.78 (1.11~2.85)

0.016

Modified Fisher grade<3

482

143 (29.7)

Modified Fisher grade≥3

213

98 (46)

Crude analysis

695

241 (34.7)

2.02(1.45~2.82)

<0.001

Multivariable analysis

695

241 (34.7)

1.72(1.11~2.67)

0.014

Point 2: There is a well-documented association between SAH (non-traumatic) and ARF which is not mentioned in the introduction (Kahn et al 2006). 

Response 2: Thanks for the reviewer’s constructive suggestion. I added the article of Kahn et al in my section of Introduction (Page 2, line 61 in red) and in the section of Reference (Page 9, line 387-388 in red)

Point 3: Lastly, there is no mention in the discussion about why this finding may be clinically relevant. Patients with severe TBI will likely have significantly decreased GCS and all of the other covariates mentioned in the study. It is unlikely that they would be observed in situations outside of an ICU and their potential for intubation would be very high regardless. It is unclear how this finding might change the management of patients, especially when this matric (Fisher's) is not typically used in this patient population.

Response 3: Thanks for the reviewer’s rigorous review and constructive comments. I added a statement of clinically relevant in the Discussion section and highlights the significance of our research (Page 7, lines 263-267 in red).

Reviewer 2 Report

It remains unclear whether there were clinical signs of brainstem dysfunction other than coma (i.e., anisocoria, hemiparesis, posturing, bilateral pupillary dysfunction) in the cohort, and to what extent such signs were correlated with respiratory failure.

Relevant literature in the field of investigation has not been cited, such as the primary description of the GCS, the WNFS coma staging (https://doi.org/10.1007/BF01773126), the Marshall classification of brain swelling after head injury, the prognostic value of brainstem lesions after head injury as assessed with MRI, the effect of prone positioning in ARDS, the results of the DECRA trial, and the results of the SYNAPSE-ICU trial.

In patients with aneurysmal SAH, the Fisher grading is not an ordinal scale with regard to outcome. Is this also true for the authors' cohort of patients with traumatic SAH? How do you justify to apply a scale that was developed to assess the severity of aneurysmal SAH to a cohort of patients with traumatic SAH, i.e., a hemorrhage with different etiology and pathophysiology?

The English language in this manuscript needs to be significantly improved.

Author Response

Point 1:  It remains unclear whether there were clinical signs of brainstem dysfunction other than coma (i.e., anisocoria, hemiparesis, posturing, bilateral pupillary dysfunction) in the cohort, and to what extent such signs were correlated with respiratory failure.

Response 1: Thanks for the reviewer’s constructive suggestion. We reviewed electronic medical records to extract data (anisocoria, hemiparesis, posturing, bilateral pupillary dysfunction) and analyzed the association between these risk factors and ARF except for hemiparesis because the missing value of hemiparesis was high (86.1%). Finally, we found only bilateral pupillary dysfunction correlated with acute respiratory failure (Table R2). After we added bilateral pupillary dysfunction into the multivariable analysis as a covariate the OR value of high-grade tSAH was stable (Table R3).

Table R2 The association between clinical signs of brainstem dysfunction and ARF

Variable

N

ARF (N/%)

Odds Ratio(95CI)

P value

No. of events/no. of patients at risk (%)

   No anisocoria

574

204 (35.5)

   anisocoria

121

37 (30.6)

Crude analysis

695

241 (34.7)

0.8 (1.09~1.22)

0.298

Multivariable analysis

695

241 (34.7)

0.77 (0.45~1.31)

0.336

No bilateral pupillary dysfunction

477

151 (31.7)

bilateral pupillary dysfunction

218

90 (41.3)

Crude analysis

695

241 (34.7)

2.63 (1.7~4.06)

<0.001

Multivariable analysis

695

241 (34.7)

2.36 (1.29~4.29)

0.005

No posturing

533

180 (33.8)

Posturing

162

61(37.7)

Crude analysis

695

241 (34.7)

1.18(0.82~1.71)

0.363

Multivariable analysis

695

241 (34.7)

1.24 (0.74~2.09)

0.415

Table R3 The association between Fisher grades and ARF in non-adjusted and adjusted group

Variable

N

ARF (N/%)

Odds Ratio(95CI)

P value

No. of events/no. of patients at risk (%)

   Low Fisher grade

481

142 (29.5)

   High Fisher grade

214

99 (46.3)

Crude analysis

695

241 (34.7)

2.06 (1.47~2.87)

<0.001

Multivariable analysisa

695

241 (34.7)

1.78 (1.11~2.85)

0.016

Multivariable analysisb

695

241 (34.7)

1.73 (1.12~2.67)

0.013

     aadjusted for adjusted for age, gender, GCS, smoking history, comorbidities, multiple injuries, characteristics of TBI, and pulmonary risk factors

      badjusted for all covariates in Multivariable analysisa plus bilateral pupillary dysfunction

Point 2: Relevant literature in the field of investigation has not been cited, such as the primary description of the GCS, the WNFS coma staging (https://doi.org/10.1007/BF01773126), the Marshall classification of brain swelling after a head injury, the prognostic value of brainstem lesions after head injury as assessed with MRI, the effect of prone positioning in ARDS, the results of the DECRA trial, and the results of the SYNAPSE-ICU trial.

Response 2: Thanks for the reviewer’s constructive suggestion. I added relevant citations in chapter 2.1 (Page 2, line 74 in red) and in the Discussion section (Page 7, line 290 in red).

Point 3: In patients with aneurysmal SAH, the Fisher grading is not an ordinal scale with regard to outcome. Is this also true for the authors' cohort of patients with traumatic SAH? How do you justify to apply a scale that was developed to assess the severity of aneurysmal SAH to a cohort of patients with traumatic SAH, i.e., a hemorrhage with different etiology and pathophysiology?

Response 3: I appreciated the reviewer’s rigorous review and constructive comments. I agree that the Fisher grading is not an ordinal scale regarding prognosis, but the incidence of symptomatic vasospasm. In our cohort of patients with traumatic SAH, the Fisher grading is not with regard to prognosis, but acute respiratory failure. The Fisher score was also used to assess the traumatic SAH in the TBI study as below:

Albertine P and et.al. The clinical significance of small subarachnoid hemorrhages. Emerg Radiol. 2016 Jun;23(3):207-11.

Point 4: The English language in this manuscript needs to be significantly improved.

Response 4: Thanks for the reviewer’s suggestion. We invited the English editors to help edit the manuscript and make the quality of the manuscript better.

Reviewer 3 Report

The study deals with subarachnoid haemorrhage (SAH) in traumatic brain injury (TBI) patients with a special reference to an acute respiratory failure (ARF). This single-site study had patient data from 771 TBI patients graded according to the severity by GCS. Occurrence and severity of SAH were assessed, the latter in Fisher scale. The presence of ARF was recorded in TBI cases with SAH. A multimodal, multi-dimensional statistical model was created to predict ARF in the patient cohort. It is reported that high Fisher grade was associated with higher incidence of ARF.

Specific points

1. The conclusions from the study are anticipated from the outset of the work, i.e. severe TBI is associated with SAH and finally, high incidence of ARF. In that sense the study design lacks innovation and is only marginally incremental to the literature of the field of intensive care.

2. Title of the paper. The title contains adjectives and terms that constrain the content of the paper and thereby, may not serve the purpose. It would be pertinent to consider revising the title to: Association between subarachnoid hemorrhage and acute respiratory failure in traumatic brain injury patients.

3. Abstract. Line 20, patient data are collected during the course of treatment, not retrospectively, the statement must reworded.

4. Page 2, lines 52-53. This sentence must be expanded to explain more closely the nature of ‘effect of traumatic SAH between brain and lung’.

5. Chapter 2.2., line 72. ‘neuroimaging features’ must be specified. Which imaging method was used and what were the features? Similarly on line 73, what were the outcome variables?

6. Page 3, lines 110-111. With reference to neuroimaging, how were these conditions verified?

7. Page 4, 5 and 6 chapters 3.2.; 3.3 and 3.5. No need to repeat the specific values for each measure, because they can be found in Tables 1 and 2 and Figure 2.

8. Page 7, Discussion section., line 226, specify CT (contrast or non-contrast?). Line 236. A confusing sentence…’Catecholamine surge can increase the release of epinephrine and norepinephrine’… Surge refers to release and both epinephrine and norepinephrine are catecholamines. Reword.

Author Response

Point 1: The conclusions from the study are anticipated from the outset of the work, i.e. severe TBI is associated with SAH and finally, high incidence of ARF. In that sense, the study design lacks innovation and is only marginally incremental to the literature in the field of intensive care.

Response 1:I appreciated the reviewer’s rigorous review and constructive comments. The original intention of this study was to assume that some molecules in the bloody cerebrospinal fluid could cause lung injury. If it is proved that severe SAH is indeed an independent risk factor for ARF, we will next explore proteomics from blood-cerebrospinal fluid to screen candidate molecules leading to acute lung injury in traumatic SAH patients.

Point 2: Title of the paper. The title contains adjectives and terms that constrain the content of the paper and thereby, may not serve the purpose. It would be pertinent to consider revising the title to Association between subarachnoid hemorrhage and acute respiratory failure in traumatic brain injury patients.

Response 2:Thanks for the reviewer’s constructive suggestion. Because our study only recruited moderate to severe traumatic brain injury patients. So I revised the title to Association between traumatic subarachnoid hemorrhage and acute respiratory failure in moderate-to-severe traumatic brain injury patients (Page 1, lines 2-3 in red).

Point 3: Abstract. Line 20, patient data are collected during the course of treatment, not retrospectively, the statement must reworded.

Response 3:Thanks for the reviewer’s constructive suggestion. We revise them in the revised manuscript (Page 1, lines 19-21 in red).

Point 4: Page 2, lines 52-53. This sentence must be expanded to explain more closely the nature of ‘effect of traumatic SAH between brain and lung’.

Response 4:Thanks for the reviewer’s suggestion. We revised the part to rewrite my study hypothesis as to the possible role of tSAH in brain-lung interactive injury (Page 2, lines 65-66 in red). We also described the problem in detail about the possible role of tSAH in brain-lung interactive injury in the Discussion part (Page 7, lines 296- 299 in red).

Point 5. Chapter 2.2., line 72. ‘neuroimaging features’ must be specified. Which imaging method was used and what were the features? Similarly on line 73, what were the outcome variables?

 Response 5:Thanks for the reviewer’s constructive suggestion. We revise them in Chapter 2.2. of the revised manuscript (Page 2, lines 85-88 in red). Neuroimaging features were evaluated by two experienced neurosurgeons based on non-enhanced CT (Page 2, lines 102-103 in red). The outcome variable was described on Page 2 lines 74-79 in red.

Point 6. Page 3, lines 110-111. With reference to neuroimaging, how were these conditions verified?

Response 6: Thanks for the reviewer’s question. The neuroimaging conditions were verified by two experienced neurosurgeons independently who were blinded to the primary endpoint described in chapter 2.3(” Exposure and covariates”) (Page 2, lines 102-103 in red).

Point 7. Page 4, 5 and 6 chapters 3.2.; 3.3 and 3.5. No need to repeat the specific values for each measure, because they can be found in Tables 1 and 2 and Figure 2.

Response 7:Thanks for the reviewer’s constructive suggestion. We revised chapter 3.2.; 3.3 and 3.5 in the revised manuscript (Page 4, lines 188-192 in red; Page 5, line 208-212 in red; Page 6, line 239-245 in red).

Point 8. Page 7, Discussion section., line 226, specify CT (contrast or non-contrast?). Line 236. A confusing sentence…’Catecholamine surge can increase the release of epinephrine and norepinephrine’… Surge refers to release and both epinephrine and norepinephrine are catecholamines. Reword.

Response 8:Thanks for the reviewer’s constructive suggestion. We revised the two ambiguous statements in the revised manuscript (Page 7, line 283 in red; Page 7, line 292-293 in red).

Round 2

Reviewer 3 Report

While the authors have addressed technical concerns indicated in the primary review, the fundamental factor of low novelty of the study prevails after revisions.

Author Response

    I appreciated the reviewer’s rigorous review and constructive comments. The original intention of this study was to assume that some molecules induced by M-STBI, are released into the bloody cerebrospinal fluid, then enter into systematic circulation and cause lung injury. This hypothesis warrants further investigation.

    This study was a pilot study. Next, we will screen candidate molecules leading to acute lung injury by proteomics from blood-cerebrospinal fluid in traumatic SAH patients.
